# A qualitative evidence synthesis exploring people after stroke, family members, carers and healthcare professionals' experiences of early supported discharge (ESD) after stroke

**Elaine O. Connor**[1,2]*, **Eamon Dolan**[2], **Frances Horgan**[3], **Rose Galvin**[1], **Katie Robinson**[1]

**1** School of Allied Health, Faculty of Education and Health Sciences, Ageing Research Centre, Health Research Institute, University of Limerick, Castletroy, Limerick, Ireland, **2** Connolly Hospital, Blanchardstown, Dublin, Ireland, **3** School of Physiotherapy, Royal College of Surgeons in Ireland (RCSI) University of Medicine and Health Sciences, Dublin, Ireland

* elaine.c.oconnor@ul.ie

## Abstract

### Objective

Early supported discharge (ESD) after stroke has been shown to generate significant cost savings and reduce both hospital length of stay, and long-term dependency. This study aimed to systematically review and synthesise qualitative studies of the experiences and views of ESD from the perspective of people after stroke, their family members, carers and healthcare professionals.

### Method

A systematic search of eleven databases; CINAHL, PubMed Central, Embase, MEDLINE, PsycINFO, Sage, Academic Search Complete, Directory of Open Access Journal, The Cochrane Library, PsycARTICLES and SCOPUS, was conducted from 1995 to January 2022. Qualitative or mixed methods studies that included qualitative findings on the perspectives or experiences of people after stroke, family members, carers and healthcare professionals of an ESD service were included. The protocol was registered with the Prospero database (Registration: CRD42020135197). The methodological quality of studies was assessed using the 10-item CASP checklist for qualitative studies. Results were synthesised using Thomas and Harden's three step approach for thematic synthesis.

### Results

Fourteen studies were included and five key themes were identified (1) ESD eases the transition home, but not to community services, (2) the home environment enhances rehabilitation, (3) organisational, and interprofessional factors are critical to the success of ESD, (4) ESD is experienced as a goal-focused and collaborative process, and (5) unmet needs persisted despite ESD.

**Data Availability Statement:** All relevant data are within the paper and its Supporting information files.

**Funding:** The authors received no specific funding for this work.

**Competing interests:** The authors have declared that no competing interests exist.

## Conclusion

The findings of this qualitative evidence synthesis highlight that experiences of ESD were largely very positive. The transition from ESD to community services was deemed to be problematic and other unmet needs such as information needs, and carer support require further investigation.

## Introduction

In 2017 there were 1.12 million incident strokes and 9.53 million people with a stroke in the European Union (EU) [1]. By 2047 it is estimated that the number of people living with a stroke in the EU will increase by 27% mainly because of population ageing and improved survival rates [1]. Stroke continues to be a major cause of disability [2, 3] with most people after stroke discharged from hospital to the home environment [4]. Despite the evidence for stroke rehabilitation [5, 6], rehabilitation service delivery may be variable in relation to quality and distribution [7] and can be fragmented with gaps in care [8].

Early supported discharge (ESD) as a model of health service delivery for people after stroke was first described in 1995 [9]. ESD facilitates a person with mild to moderate stroke to be discharged from the acute hospital environment earlier than conventional care to continue their rehabilitation within their home with rehabilitation from members of a multi-disciplinary team (MDT) [10, 11]. Average length of hospital stay before ESD in the United Kingdom is a median of 6.9 days (IQR 2.9–18.8) [12]. International best practice guidelines endorse ESD as a form of rehabilitation that should be offered to those following a stroke that are suitable to participate [13–16]. A Cochrane review of ESD for people after stroke demonstrated a reduction in the length of hospital stay by 6 days [CI -3 to -8 days] and a reduction in long term dependency and admission to long term care [17]. A more recent review also demonstrated improved activities of daily living (ADL) scores for those receiving ESD compared to those receiving usual care [18]. ESD has also been proven to be cost effective [19–21].

In recent years, qualitative research with people after stroke has been synthesised on topics including return to work, physical activity, self-management, medication management and living with a stroke [3, 22–24]. These syntheses have illuminated common dimensions of the stroke experience including the need for bridging between the clinical and employment spheres, the experience of social isolation, the burden of carer work and the need for occupational adaptation after a stroke [3, 22–24]. The James Lind Alliance [25] prioritised areas in stroke rehabilitation whereby evidence is currently not up-to-date or reliable. These areas include researching people after stroke and carers experiences of the stroke pathway, including intensity of rehabilitation [25].

Several qualitative studies reporting the experience of ESD from the perspective of people after stroke, carers and healthcare professionals (HCPs) have been published. This body of work was previously reviewed and synthesised using a narrative synthesis approach to analysis by Osborne and Neville [26]. Two main themes were identified in their synthesis: psychosocial aspects of ESD and the logistical components of ESD. Osborne and Neville's [26] database searches were conducted in October 2018, however no papers published after 2016 are included in their review. Furthermore, only three databases were searched (MEDLINE, CINAHL and Embase) and their database searches yielded only 147 records after duplicates were removed indicating that their search strategy was very focused, with six studies included after full text review.

We aimed to expand on the findings of this review and offer a more nuanced account of participant experiences of ESD by increasing the number of databases and search terms used, including both peer reviewed studies and theses and by applying Thomas and Harden's [27] three stage approach to thematic synthesis. The primary aim of this synthesis is to explore the totality of evidence regarding the experiences and views of people after stroke, family members, carers and HCPs of an ESD service to inform future service developments and research.

## Methods

### Study design

A qualitative evidence synthesis approach was used to address the research question: what are the experiences and views of people after stroke, family members, carers and healthcare professionals of Early Supported Discharge (ESD)? Qualitative evidence synthesis, also known as qualitative systematic review, offers a means of aggregating or integrating findings from multiple qualitative studies [28]. The conduct and reporting of this qualitative evidence synthesis has adhered to ENTREQ guidelines and the completed checklist can be found in S2 File [29]. The original protocol can be accessed using the Prospero database (Registration: CRD42020135197).

### Identifying papers for inclusion and exclusion

**Types of studies.**   We included primary studies that used recognised methods of qualitative data collection and data analysis. Studies that used mixed methods where qualitative data could be extracted were also included.

**Topics of interest.**   Studies reporting the experiences or views of people after stroke, family members, those involved in managing or commissioning or delivering health or social care services on ESD for people after stroke were included. By ESD, we mean a model of care whereby people with mild to moderate stroke have an accelerated discharge from the acute setting to the home environment to continue their rehabilitation at home as an alternative to conventional hospital-based care [9, 30–32]. Articles were excluded if they were pertaining to ESD but not specific to stroke, or described healthcare services for people after stroke but were not specific to ESD.

**Search selection.**   A systematic literature search of eleven databases (CINAHL, PubMed Central, Embase, MEDLINE, PsycINFO, Sage, Academic Search Complete, Directory of Open Access Journals, The Cochrane Library, PsycARTICLES, and Scopus) was conducted from 1995 to January 2022. All searches were conducted by the first author (EOC). The search strategy included three key concepts; "early supported discharge", "stroke", and "qualitative research" in conjunction with MeSH terms. The first five pages of Google Scholar results were also searched using the three key concepts. The search string can be found in S1 File. The reference lists of included articles were hand searched. The search was restricted to English language publications from 1995 as the first studies pertaining to ESD in this population were published in 1995. Peer reviewed publications and theses were included.

**Selection procedure.**   All references from the database searches were imported into End-Note X9 by EOC with duplicates subsequently removed. Two authors (EOC and KR) independently screened the titles and abstracts of the identified records for eligibility. A third author (RG) was consulted where consensus was not achieved. Two reviewers (EOC and KR) independently read each full text article identified for inclusion in the synthesis. A third reviewer (RG) was consulted to read any full text article when consensus could not be reached.

**Critical appraisal and data extraction.**   We appraised the quality of the included studies using the 10-item Critical Appraisal Skills Programme (CASP) checklist for qualitative studies

**Table 1. Quality appraisal of included studies.**

| Study name | CASP Criterion 1<br>Clear statement of aim | CASP Criterion 2<br>Qualitative methodology appropriate | CASP Criterion 3<br>Appropriate research design | CASP Criterion 4<br>Sampling | CASP Criterion 5<br>Data collection | CASP Criterion 6<br>Relationship between researcher and participant considered | CASP Criterion 7<br>Ethical issues | CASP Criterion 8<br>Data analysis and rigour | CASP Criterion 9<br>Clear statement of findings | CASP Criterion 10<br>Value of research |
|---|---|---|---|---|---|---|---|---|---|---|
| Chouliara et al. (2014) | Yes | Yes | Unclear | Yes | Yes | No | Yes | Unclear | Yes | Yes |
| Cobley et al. (2013) | Yes | Yes | Unclear | Yes | Yes | No | Yes | Unclear | Yes | Yes |
| Collins et al. (2016) | Yes | Yes | Yes | Unclear | Yes | No | Yes | Yes | Yes | Yes |
| Fisher et al. (2021) | Yes | Yes | Yes | Yes | Yes | Unclear | Yes | Yes | Yes | Yes |
| Hitch et al. (2020) | Yes | Yes | Yes | Yes | Yes | No | Yes | Yes | Yes | Yes |
| Kylén et al. (2021) | Yes | Yes | Yes | Yes | Yes | No | Yes | Yes | Yes | Yes |
| Lou et al. (2017) | Yes | Yes | Unclear | Yes | Yes | No | Yes | Yes | Yes | Yes |
| Moule et al. (2011) | Yes | Yes | Unclear | Yes | Yes | No | Yes | Yes | Yes | Yes |
| Rider (2015) | Yes | Yes | Unclear | Yes | Yes | Unclear | Yes | Yes | Yes | Yes |
| Rochette et al. (2021) | Yes | Yes | Yes | Yes | Yes | No | Yes | No | Yes | Yes |
| Shiggins (2017) | Yes | Yes | Yes | Yes | Yes | Yes | Yes | Yes | Yes | Yes |
| Taule et al. (2015)b | Yes | Yes | Yes | Yes | Yes | Yes | Yes | Yes | Yes | Yes |
| von Koch et al. (2000) | Yes | Yes | Unclear | Yes | Yes | No | Yes | Unclear | Yes | Yes |
| Wohlin Wottrich et al. (2007) | Yes | Yes | Yes | Yes | Yes | No | Yes | Yes | Yes | Yes |

[33]. Two reviewers (EOC and KR) independently appraised the included studies with a third reviewer (RG) involved where any differences of opinion arose. The CASP checklist for each of the included studies can be found in Table 1. The CASP checklist is endorsed by the Cochrane Qualitative and Implementation Methods Group [33] and is commonly used for quality appraisal in health-related qualitative evidence syntheses [34]. Although studies were not excluded based on CASP scores, critically appraising studies served to evaluate and allow discussion on the credibility of findings within the research team which influenced the process of analysis [35]. A customised template, Table 2, was used to extract the descriptive characteristics of each study which included the number and characteristics of participants, location of study, methodology, data collection and key research aims.

**Thematic analysis.** Findings from included studies were imported into NVivo Version 12. Thematic synthesis is a commonly used approach for a qualitative evidence synthesis [36] that enables transparency with accessible outcomes [37, 38] and attempts to go beyond the original data to seek a fresh interpretation of the phenomena under review [39]. Thomas and Harden's approach to thematic synthesis involving a three-stage process was undertaken [27]. In the first phase of analysis all data were coded line by line according to its meaning and

**Table 2. Characteristics of included studies.**

| No | Authors | Sample Characteristics | Gender Breakdown | Age | Approach | Data Collection | When interviews were conducted | Research Aim |
|---|---|---|---|---|---|---|---|---|
| 1 | Chouliara et al. 2014 United Kingdom | 35 key informants —practitioners, managers and commissioners | Unknown | Not listed | Thematic analysis | Face to face semi-structured interviewers by one of two researchers and lasting approximately 45 minutes | Not stated | To explore the perspectives of healthcare professionals and commissioners working with a stroke early supported discharge service in relation to: (1) the factors that facilitate or impede the implementation of the service, and (2) the impact of the service. |
| 2 | Cobley et al. 2013 United Kingdom | 27 patients, 15 carers—13 carers female—all carers spouses however those receiving ESD were 19 patients and 9 carers | Unclear with stroke patients and only mentions carer breakdown | Mean age of patients—69.85 ±13.42 years, mean age of carers— 72.79 ± 14.10 years | Thematic analysis | Semi structured interviews ranging from 30–45 minutes | 1–6 months following discharge from hospital | To investigate patients' and carers' experiences of early supported discharge services and inform future early supported discharge service development and provision. |
| 3 | Collins et al. 2016 Ireland | 4 patients | 2 Female, 2 Male | 61–81 years | Interpretative phenomenological analysis | Semi–structured interviews lasting 45–90 minutes | Two weeks and three months post discharge from the early supported discharge service | To explore the experience of early supported discharge from the perspective of stroke survivors in Ireland. |
| 4 | Fisher et al. 2021 United Kingdom | 117 = healthcare professionals 30 = patients | Healthcare professionals– unknown Patients– 10 (F) and 20 (M) | Healthcare professionals– unknown Patients—32– 88 years (mean of 65.9 years) | Healthcare professionals' interviews were analysed following an iterative, retroductive process moving between deductive and inductive phases. Patient interviews were analysed using the six stages of reflexive thematic analysis recommended by Braun and Clarke | N = 35—Semi-structured one-to-one interviews with up to 8 NHS staff (staff informants at senior management, service lead and commissioning level at each early supported discharge site N = 82–2 group interview sessions at each site with early supported discharge team interviews Semi-structured interviews with patients ranged from 24–70 minutes | Healthcare professionals– September 2018 – August 2019 Early supported discharge patients– November 2018 – November 2019 | Healthcare professionals—To obtain a better understanding of the interaction between contextual influences, core intervention components and the reasoning and actions of staff members involved with early supported discharge delivery. Early supported discharge patients —To explore their views and experiences of early supported discharge services. |

*(Continued)*

**Table 2.** (*Continued*)

| No | Authors | Sample Characteristics | Gender Breakdown | Age | Approach | Data Collection | When interviews were conducted | Research Aim |
|---|---|---|---|---|---|---|---|---|
| 5 | Hitch et al. 2020 Australia | 23 healthcare professionals participated in focus groups or interviews. 111surveys were received from healthcare professionals | Unknown | Not listed | Priori thematic analysis | Semi-structured interviews = 7—Medical ST, Neuropsych, OT and Physio. Focus groups = 16—Admin, Nursing, ST, Psych, OT and Physio. | Six month period | To describe staff perceptions of the trial of an early supported discharge model of care for stroke survivors at a large metropolitan public hospital in Australia. |
| 6 | Kylén et al. 2021 Sweden | 17 patients | 8 (F) and 9 (M) | 34–90 (f)/46-81 (m) years | Inductive qualitative content analysis | Semi-structured interviews ranging from 20–30 minutes | August 2019 to January 2020 (conducted approximately 3 months post-stroke) | To explore how the environment was integrated into rehabilitation at home from the perspective of patients after a stroke. |
| 7 | Lou et al. 2017 Denmark | 22 patients, 18 partners | 22 patients ESD—7/22 (F), 15/22 (M). 18 partners 4 (M) and 14 (F) | N = 22 average age 65 years (f) and 70 years (m) | Thematic analysis | Semi-structured qualitative interviews ranging from 30–60 minutes | 3–6 weeks following stroke | To investigate how mild stroke patients' and their partners' experience and manage everyday life in a context of early supported discharge. |
| 8 | Moule et al. 2011 United Kingdom | 6 team members, 4 external stakeholders | Unknown | Not listed | Thematic analysis | Semi-structured interviews lasting approximately 1 hour | October 2009 – February 2010 | The aim of the research was to address the question: "How did the ESD team members and external stakeholders experience the service implementation process?" |
| 9 | Rider 2015 United Kingdom | 8 patients, 5 carers | 5 (F) and 3 (M)—patients; 3 (F) and 2 (M)–carers (8 patients and 5 carers) | 47–86 years patients only. No age given for carers | Inductive qualitative content analysis | Semi-structured interviews lasting between 15–57 minutes | Two distinct chronological time points for interviews—3 weeks post discharge home and within 2 weeks of discharge from the early supported discharge team between April and October 2014 | To investigate patients' and their carers' experiences after being discharged from hospital home with the support of a five-day a week early supported discharge team. |

(*Continued*)

**Table 2.** (Continued)

| No | Authors | Sample Characteristics | Gender Breakdown | Age | Approach | Data Collection | When interviews were conducted | Research Aim |
|---|---|---|---|---|---|---|---|---|
| 10 | Rochette et al. 2021 Canada | 90 participants however n = 29 in the context of early supported discharge | 21(F) and 5(M)—missing data | 3 = 45 years and less, 8 = 46–55 years, 7 = 56–65 years, 6 = 66–75 years, 3 = 76 years and more | Thematic analysis | 320 comments from 90 participants were grouped under 6 themes | Not stated | To describe their (relatives) perceptions of the quality of the services they received in the context of early supported discharge, in- and out-patient rehabilitation services. |
| 11 | Shiggins 2017 United Kingdom | 9 persons with aphasia, 8 healthcare professionals | 3 (F) and 6 (M)–persons with aphasia; 7 (F) and 1 (M) healthcare professionals | 52–92 years persons with aphasia, only years of experience given for healthcare professionals | Thematic analysis | 8 healthcare professionals—semi-structured topic guided interviews—33–60 minutes, 9 persons with aphasia—one to one semi-structured topic guided interviews—15–71 minutes | Recruitment over 15 month period beginning with healthcare professionals from December 2013 –February 2015; when the recruitment target for participants with aphasia was reached | The overall aim of this study is to explore opportunities for functional communication (re)learning in the context of routine early supported discharge rehabilitation (rehabilitation conducted in day-to-day practice) between healthcare professionals and people with aphasia. |
| 12 | Taule et al. 2015b Norway | 8 patients | 4 (F) and 4 (M) | 45–80 years | Interpretive description, systematic text condensation and coping theory | Qualitative interviews in the context of a randomised controlled trial | 6–8 months post stroke | To explore mild-to-moderate stroke survivors' experiences with home rehabilitation after early supported discharge from hospital. |
| 13 | von Koch et al. 2000 Sweden | 41 patients (quantitative) 6 therapists (qualitative) | 19 (F) and 22 (M) patients (quantitative) 6 therapists = female (qualitative) | 70.8/72 (49–86 range) patients (Quantitative) 20–45 years therapists (Qualitative) | Qualitative—Narrative manner, a step-by-step procedure was adopted in which each interview was analysed separately | Semi structured interviews with each therapist | Not stated | To describe the content of a programme involving early hospital discharge and continued rehabilitation at home after stroke. |
| 14 | Wohlin Wottrich et al. 2007 Sweden | 13 members of a multi-professional outreach team on stroke rehabilitation in the patients' home | 8 (F) and 5 (M) | 23–51 years | Empirical Phenomenological Psychological (EPP) method | Interviews conducted by the first author lasting 30 minutes | 1 week after the person had been discharged from the home rehabilitation program | To identify the meaning of rehabilitation in the home environment after stroke from the perspective of members of a multi-professional team. |

content by two members of the team independently (EOC & KR). Across all 14 studies there were extensive descriptions of the broader stroke experience such as the experience of a changed body however only findings related to ESD were coded in keeping with our research question. At this stage the researchers remained close to the results of the included studies. This process enabled translation of concepts from one study to another. In the second phase of analysis, discussion between the researchers facilitated the grouping of codes generated in the first stage into a hierarchical structure and subsequently these groupings were named as descriptive themes. In the final phase of analysis, descriptive themes were reviewed by two members of the research team (EOC & KR) and discussed in light of the research question to develop analytic themes which went beyond a descriptive account of findings presented in the included studies.

## Findings

### Study identification

The initial search yielded 4156 articles with 877 duplicate articles removed. 3259 articles were excluded because they did not meet the inclusion criteria. From the remaining twenty articles, full texts were obtained and eligibility was determined by the three authors (EOC, KR and RG). Fourteen articles were included in the final review. The selection process is outlined in Fig 1. Eleven studies were located through database searches, one study through Google Scholar [40]; one study through the protocol of the study initially found in the database search [41] and one study retrieved from a conference abstract identified again in the database search [42].

### Descriptive characteristics of the included studies

Table 2 outlines the details of the 14 included studies. All included studies were conducted in high income countries as indicated by The World Bank [43]. Six studies were conducted in the United Kingdom [40, 42, 44–47], three studies were conducted in Sweden [41, 48, 49] and one study was conducted in Denmark, Ireland, Canada, Australia and Norway respectively [50–54]. One study did not analyse data collected via interview, instead analysing responses to open ended survey questions [52]. Five studies focused on HCPs and/or those managing or commissioning health services [45, 46, 48, 49, 53], two studies included both people after stroke and HCPs [40, 47], three studies included both people after stroke and carers [42, 44, 50], three studies only included people after stroke [41, 51, 54] and one study focused solely on carers [52].

The 14 studies in this review reported on the experiences of 390 participants in total. The study with the smallest number of participants was Collins and colleagues (n = 4) [51] while the study with the largest number of participants was Fisher and colleagues (n = 147) [40]. The age range of participants with stroke ranged from 32 years [40] to 92 years [47]. All included studies reported some dimension of the experiences or views of people after stroke, family members, those involved in managing or commissioning or delivering health or social care services on ESD for people after stroke. The aim of five included studies aligned very closely with the focus of this synthesis. These five studies explored the experience of ESD from the perspective of the person after stroke and/or carers [40, 42, 44, 51, 54]. Fisher and colleagues [40] report articulated a more focused aim for exploring HCPs perspectives, namely, to understand the interaction between contextual influences, core intervention components and the reasoning and actions of ESD team members. Similarly, Chouliara and colleagues [45] aimed to explore perspectives on implementation factors and the impact of the ESD service and Moule and colleagues [46] explored the ESD service implementation process. In the study by Hitch and colleagues [53], the aim was to describe HCPs perceptions of the trial of an ESD

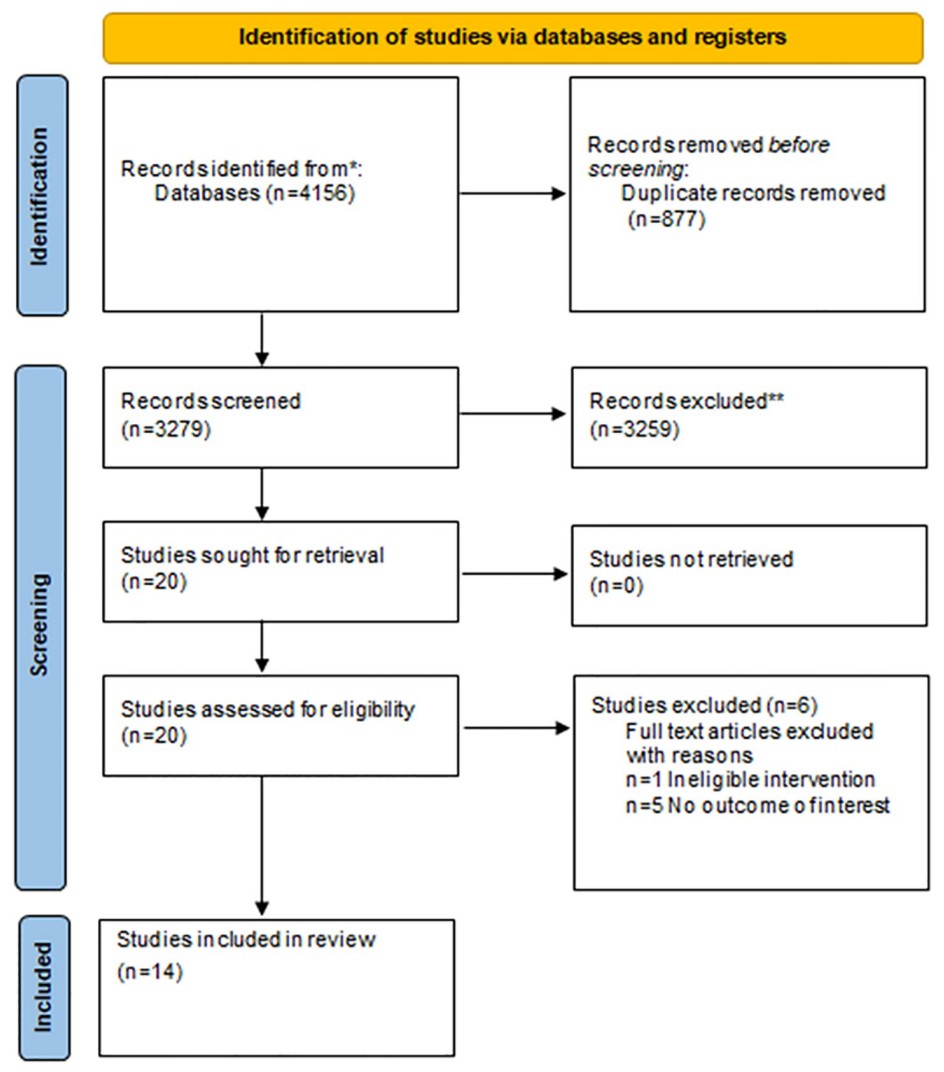

**Fig 1. PRISMA flow diagram.**

model of care in Australia. Von Koch and colleagues [48] aimed to describe the content of an ESD program whereas Wohlin Wottrich and colleagues [49] aimed to identify the meaning of rehabilitation in the home environment from the perspective of healthcare providers. Four studies articulated a more focused or refined aim related to the views or experiences of people after stroke or carers [41, 47, 50, 52]. Kylén and colleagues [41] explored how the environment was integrated into ESD from the perspective of people after stroke. Lou and colleagues [50] investigated how those with a mild stroke and their partners' experience and manage everyday life in the context of ESD. Shiggins [47] explored opportunities for functional communication (re)learning in the context of ESD. Finally, Rochette and colleagues [52] aimed to investigate carers' perceptions of the quality of services they received.

## Quality appraisal

Findings from the quality appraisal are presented in Table 1. All 14 studies were clear in the aims of their research, the methodology was appropriate and there was a clear statement of the

findings. For 12 studies, the relationship between the researcher and the participants did not appear to be considered. The appraisal of the studies was conducted by two authors (EOC and KR) and no consensus was required.

## Five key themes were identified

**Theme 1: ESD eases the transition home, but not to community services.** Across the included studies, ESD was described as easing the transition from hospital to home, however the transition to community services at the cessation of ESD was more problematic. Prior to discharge people after stroke had anxieties about going home [51]. Although HCPs asked about the home environment before discharge and potential barriers [41] some people after stroke struggled to predict the challenges they would face at home or to know or express their wishes for rehabilitation [54]. Limited knowledge of ESD [40, 51] or involvement in decision making about ESD prior to discharge [41] was reported by people after stroke.

*"Before I actually came out of the hospital, before coming home, I panicked slightly, and I thought going through my mind, how am I going to get around with the walker?. . . "I'll never manage, what am I going to do?"'*

[51]

The experience of discharge home with ESD was described as *"seamless"* by the majority (16/19) of people after stroke in the study by Cobley and colleagues [44], with the intensity of ESD (up to four visits per day, seven days per week for a duration of six weeks) provided a sense of security [44]. Similarly, having an ESD visit on the day of discharge [42] or shortly after discharge [50] was valued by people after stroke for "continuity of therapy, support and assistance with immediate problems encountered in the early days post discharge" [40].

Although they were discharged early, several studies reported that people after stroke felt the timing of discharge was appropriate [42, 50]. While the hospital to home transition was eased by ESD, in contrast, discharge from ESD to community services was often described as problematic. More than half of the people after stroke in one study [40] were concerned about what was going to happen after the initial 6 weeks of the ESD service ended.

*"I don't know. I. . . I was just worried after this 6 weeks. What happens after this 6 weeks. Because you know you have these people come four times a week and then all of a sudden . . .. what's going to happen? And they said there is another plan [inaudible] . . . but it could take time".*

[40]

Carers in another study were also concerned about the transfer to community therapy, with concerns voiced around reduced therapy intensity, getting to know new HCPs and break in therapeutic equipment use [42]. Those studies that included details on transition to community services after ESD highlighted problems including challenges securing social care input, disjointed transition to other services and a lack of community stroke services [44, 45]. In contrast, in one study [50], people after stroke who continued with municipal rehabilitation after ESD in Denmark generally reported the transition to be unproblematic. HCPs, in the study by Wohlin Wottrich and colleagues [49], also described anxieties at the cessation of ESD if goals had not been achieved. It was suggested by HCPs in one study that a combined service with ESD and community stroke team components could meet the needs of those with further rehabilitation potential at the cessation of ESD [40].

**Theme 2: The home environment enhances rehabilitation.** HCPs reported that people after stroke prefer to get home from hospital [45, 46, 49, 51, 53] and were eager [47] and happy to return home [44]. People after stroke reported positive feelings associated with being at home such as feelings of safety, security, familiarity, comfort and relaxation [47, 50, 51, 54]. These experiences at home were contrasted with the hospital atmosphere by a participant with aphasia (PwA);

*"PwA 3 pantomimed marching to depict the atmosphere in the hospital. '(points forward towards a cup of tea on the table) You can have a cup of tea yea yea Like we are now Yes (smile)' (PwA 3)"*

[47].

Participants noted that travel to outpatient or hospital appointments in the first weeks after discharge would have been difficult due to various challenges including wheelchair dependency, cost, limited public transport, driving cessation and fatigue [40, 42, 47].

*"They said, 'oh you'd probably have to go there for physio'. And I thought, God how am I going to get there? You know, I'd have to have a taxi every day. And I thought, God I can't afford that. And then when they said about these coming in, I was like 'oh that's brilliant'. You know, because it's like £6 each way in a taxi. I wouldn't be able to afford that every. . . four times a week".*

[40]

The home environment promoted better sleep [42, 50], recovery [41, 42, 50] and a return to pre-stroke activities and routines thus contributing to a sense of normality, continuity [49, 51] and adjustment post stroke [41, 47]. Therapy at home helped identify discrepancies between *"the desires and abilities of the patient on the one hand and the environmental demands and expectations on the other"* [48] which could make some difficulties apparent such as cognitive difficulties [42] and thus supported problem solving [45, 47, 48, 49]. This was described by participants in the study by Chouliara and colleagues [45] as an *"ecologically valid appraisal of patients' difficulties"*. People after stroke found it meaningful to practise and recover in their everyday surroundings [50] and described ESD as personalised or tailored to their individual needs [40, 44, 45, 51] in contrast to hospital services. HCPs also described rehabilitation at home as more meaningful [47, 53]. The home environment also aided rehabilitation by providing a greater number of and a wider range of task practice opportunities in the home and local area than hospital [41, 42, 44, 47] for example, *"carpet-bowls and exercising the dog"* [42].

*"I wouldn't of wanted to go out on my own but the ESD Therapy Assistant Practitioner took me" "I took her arm and we went to the shop but later on I went out walking and did a bit on my own".*

[42]

In one study focused on people with aphasia (PwA), HCPs felt naturalistic conversation was enhanced by the home environment as the PwA was more comfortable, the home environment helped establish rapport and gave clues about the person, thus acting as a scaffold and springboard for conversation [47]. Most studies, bar one [41] described extensive use of the home environment by HCPs. The researcher in one study observed [47] that occupational therapists (OTs) and physiotherapists (PTs) used meaningful activities and the home

environment more than speech and language therapists (SLTs) during ESD, with the SLT *"standardised and formal materials were often brought with the HCPs to the person's home, such as laptops and picture cards. Fewer materials readily available in the home were used compared to OT and PT activities"*.

**Theme 3: Organisational and interprofessional factors are critical to the success of ESD.** This theme relates to various factors that influenced the success of ESD. Key factors included the need for clarity on the ESD model of care, the need for some flexibility in the application of ESD criteria, the need for advance planning of discharge from ESD, the need for person-centred ESD scheduling, and the need for ESD HCPs to work effectively as an inter-professional team. Clarity on the model of care was felt to be important to the success of ESD. While HCPs felt good communication with commissioners was important to establish the position of ESD in the stroke care pathway [40], commissioners themselves reported lack of clarity about the position of ESD in the pathway and lack of clarity about its effectiveness [45].

> *"To be honest I am bit foggy about where Early Supported Discharge sits alongside intermediate care and re-enablement and how these are married up"*.

> [45]

In contrast, HCPs felt ESD aligned with the provision of best care [53] and they also *"recognized that ESD addressed the broader agendas on patient choice and patient goals"* [46]. In developing the model of care, the pathway needed to be agreed with the various health and social care stakeholders [46]. The stroke pathway could be hampered by delays at transfers of care [54], lack of ESD slot availability, staff shortages, inadequate care packages [40], impasses over payment for social care [46], tensions between ESD and community rehabilitation teams [46] and challenges with the transfer to community services after ESD [40, 42, 44, 45].

> *"For instance, staff at site D reported that they could offer the recommended intensity to a maximum of 19 patients, but there were 30 patients in their caseload at the time of the interviews"*.

> [40]

To increase therapy intensity without further stretching capacity, HCPs trained family members and care providers, implemented telerehabilitation and promoted self-management [40, 45]. In several studies, HCPs emphasised that ESD criteria are important but needed to be implemented with flexibility [40, 45, 47] for example delaying discharge from ESD or taking on some people after stroke who didn't meet eligibility criteria. Challenges with accessing community services or care packages and service fragmentation were reasons why flexible criteria were needed.

Flexibility was also needed to meet the needs of a 'new type' of person after stroke, those post thrombolysis and thrombectomy, who might not meet ESD criteria in terms of physical function needs but who might have potentially hidden issues (e.g. cognition and mood) and different needs (e.g. vocational) [40]. HCPs felt discharge from ESD services should be actively managed and prepared, with timely communication to avoid an abrupt end [40, 48]. Longer spaces between ESD visits preceded discharge [40, 48] and people after stroke appreciated this gradual reduction in visits, joint handover visit by ESD and community staff, and knowledge that the ESD team could be contacted post-discharge [42, 50].

Person-centred scheduling of ESD was supported by participants. In two studies people after stroke valued having ESD timetables [42, 47] to allow planning for formal care visits,

social and leisure time, and avoid confusion. Similarly, HCPs felt the intensity of rehabilitation needed to be person-centred rather than dictated by a guideline as *"patient feedback often substantiated this view, with patients requesting the time and space to 'feel at home' and resume some of their activities"* [40].

Interpersonal and interorganisational communication was described as essential to ESD success. A proactive approach to develop communication, trust and rapport between the ESD team and the acute setting, especially in the early days of ESD development to ensure referrers understood the screening and referral process and appreciate the quality of the ESD service was considered important [40, 45, 46, 53]. To support communication between ESD teams and hospital staff, meetings, common training events and working arrangements, such as staff rotations across services [45], shadowing colleagues across services [40], and in-reach by ESD HCPs to the acute setting [49] were described as beneficial.

Inadequate data sharing and lack of joined up or shared IT system was a barrier to communication between ESD and the acute hospital [40, 45]. This was contrasted with the positive experience of using SystmOne in one study where teams could access each other's notes [40]. Meetings across the *"entire pathway"* such as regular service management meetings, or quality improvement meetings were described as promoting *"clarity on the availability and role of each service and awareness of the pressure points in the pathway"* thus streamlining communication leading to a smoother and safer handover between services [40]. A case manager [48] or ESD co-ordinator role [53] was operationalised in some teams with perceived benefits in terms of championing the service and working across service boundaries.

Four studies discussed the benefits of rehabilitation assistants (RAs) or assistant practitioners as part of the ESD team [40, 45–47]. Generic RAs with an interdisciplinary skillset in the team promoted a holistic and interdisciplinary rehabilitation ethos. Benefits of RAs included freeing highly skilled HCPs from repetitive everyday exercises to focus on *"the more specialist elements of rehabilitation"* [45], thus *"maximising the service capacity and cost-effectiveness"* [40].

*"To be honest, if patients really needed a physiotherapist every single day for the whole of the 6 weeks, that would be hard to do. But actually, we've upskilled our assistants to a level now where once we've got our plan and we're happy with that plan and to deploy them, and they can go. It depends on their need".*

[40]

Effective multi-disciplinary team (MDT) working was described as important for ESD service delivery [48, 53] and was enhanced by interdisciplinary practices, team meetings, a shared office base to support communication, good ESD team leadership [40, 46] and collegiality [40]. In one study, ESD HCPs valued having a stroke-specialist nurse in the team to address medication queries, information gaps around medical issues, facilitate holistic care and improving efficiency [40]. One team used the term *"nerapists"* as a reference to nurses' and therapists' shared skillset, while interdisciplinary working was valued and embraced, they emphasised the need for balance between specialism and interdisciplinarity to avoid duplication [40].

*"You play every role. You play an occupational therapist, you play a speech and language therapist, you play physio, don't you, you don't just go in and do one thing that is physio. You talk about how they're feeling, what appetite, sleep is like, how they're managing their everyday tasks, how their speech is. You cover all bases".*

[40]

**Theme 4: ESD is a goal focused and collaborative process.** Across the studies, ESD was described as a process of discovering problems, setting goals, resolving problems and evaluating outcomes [48, 49]. People after stroke described goal setting as a collaborative process and identified that therapy reflected their goals or needs [42, 45, 47]. Similarly, HCPs described focusing goals on the interests and priorities of people after stroke [48, 49] and asking them to share life stories to understand their priorities to inform future therapy and goal setting [49]. A small number of contrasting experiences were reported such as people after stroke describing being informed rather than involved in the planning of their ESD [41] and non-person-centred goal setting [54]; *"They just had a plan of returning me back to work. It was their goal. When they repeated that every time we met, then I started to cry I think, every time"*. [54]

Across the studies, descriptions of positive relationships between HCPs and people after stroke were common. HCPs and people after stroke reported friendship type relationships [47, 51]; *"They were such nice girls, I looked forward to the camaraderie we had; we had great chats and craic"* [51]. Across the studies, participants described people after stroke being more empowered in their own homes and taking a more active role in rehabilitation or equal role in the HCP-people after stroke relationship [40, 47, 48, 50, 51]. People after stroke appreciated HCPs reassuring, friendly, positive attitude towards them, and celebration of success [47]. However, they wanted ESD HCPs to communicate realistic hope, delicately balancing optimism for recovery with realism and the delivery of tailored feedback on their outcome and performance in tasks rather than generic praise [47, 54].

*"They [the municipal healthcare team] really came and stayed here and did something. They showed faith in positive development and supported me in that. It's important to convey that recovery can still happen, although the progress is slow"*.

[54]

Shiggins [47] observed detailed, accurate and explicit feedback about performance from HCPs, however, untailored feedback was also observed.

*"PwA 7 was also aware when a HCP was providing her with untailored feedback, when at the end of a session she began to tut and screw up her face in response to HCP 13 telling her that she had 'done good', when in fact she had little success in the session and was aware of this"*.

[47]

People after stroke wanted HCPs to listen, be empathetic [47, 50, 54], have patience [42] and to allow the person after stroke to be equal in decision making [54]. HCPs accounts of the relationship with people after stroke mirrored what participants reported they wanted to a large degree. HCPs reported dialogue and collaboration with people after stroke as an important vehicle for rehabilitation [48].

*"In the hospital, this big institution where you are an authority in a white coat, the patient submits himself to you and wants you to help him and make him well. But at home I think it's more like you discuss the patient's problems and co-operate with him to find solutions"*.

[48]

Observations of ESD reported mixed findings on the extent of collaboration between HCPs and people after stroke. Genuine collaboration was observed [47]. However, HCPs were more in control of sessions and conversation than people after stroke such as using specific phrases

to signify the end of an activity, they also exerted control through their positioning of and control of rehabilitation materials [47].

**Theme 5: Unmet needs persisted despite ESD.**   People after stroke reported many important improvements due to ESD [40, 44] such as increased mobility, reduced fatigue, a return to driving [42], speech improvements [47] and ESD supported adjustment to disability post stroke [40]. Despite these positive outcomes a number of unmet needs were identified across studies including information needs and carer support needs. People after stroke and carers reported a lack of knowledge about stroke, causes, secondary prevention and medication [42, 44, 50, 54]. Generic information not tailored to their specific needs, was not valued [44].

*"You read the pamphlets, the leaflets and things, what to look for with strokes, but I mean the thing is, a lot of the things in there weren't applicable".*

[44]

Both people after stroke and carers reported difficulties accessing information concerning benefits, allowances, community supports and resources [44, 52]. Participants valued HCPs knowledge of *"the wider stroke pathway and health community and relied on them to signpost to other agencies for support"* [40] and a link with other services [42, 50]. HCPs described acting on behalf of people after stroke to retrieve information or convey messages. *"We act like a kind of an ombudsman for the patient. We make it easier for the patients and you assist them in finding the right authority for their problems"* [48].

A further area of unmet need related to the needs of carers, who often reported being insufficiently supported by ESD. Carers described feeling they were thrown into the caring role and described responsibility for the person after stroke as a shock [42, 44]. They described the return home of the person after stroke as a major transition for them personally and taking on jobs that the person after stroke used to do [44] in addition to providing physical care, and new activities such as administering medication. In some cases, they were unable to leave the person after stroke alone due to safety concerns [42] and struggled with emotional changes such as continuous crying [44]. These changes were described as exhausting [42, 44].

*"Since he's come home, I've not really gone out very much. Normally I would just go out and do whatever, but I haven't been able to do that since he's come home from hospital".*

[44]

HCPs also suggested the need to better support families and, in some cases, presented examples to identify and respond to carers needs [40, 53].

*"That's the one piece of feedback I get from every single family member, I didn't realise how hard this was going to be and I didn't realise what it meant to be caring for them".* [53] Carers suggested a *"special session for relatives of people who have had a stroke"* [52], longer ESD sessions so they could get out of the house and have some respite [44] and more gradual transitions of care [52].

Other unmet needs were reported including insufficient emotional or psychological support for people after stroke [40, 54], insufficient attention to cognitive issues, and activities valued by people after stroke [41] such as social activities [54] and work or vocational needs [50].

## Discussion

### Statement of principal findings

This study aimed to systematically review and synthesise qualitative studies of the experiences and views of ESD from the perspective of people after stroke, their family members, carers and HCPs. We identified five key themes; ESD eases the transition home, but not to community services (theme 1), the home environment enhances rehabilitation (theme 2), organisational and interprofessional factors are critical to the success of ESD (theme 3), ESD is a goal focused and collaborative process (theme 4) and unmet needs persisted despite ESD (theme 5).

People after stroke reported largely positive experiences of ESD. This aligns with findings from the Riksstroke registry in Sweden which showed that those with first ever stroke who received ESD (n = 1495) were more satisfied with rehabilitation after discharge, reported less depression and better function compared to controls (n = 28,737) [55]. Our findings on important improvements reported by people after stroke following ESD such as improved mobility and the ability to resume valued activities also mirrors findings from systematic reviews of the effectiveness of ESD after stroke which have found improvements in participants' activities of daily living [ADL] scores [18] and extended activities of daily living scores [17] in favour of ESD when compared to usual care.

One key finding of this synthesis is that the home environment was reported to be an optimal environment for rehabilitation. Both HCPs and people after stroke reported greater empowerment of the person after stroke in the home environment. This is important because other studies reported that a lack of collaboration and teamwork between people after stroke, carers and HCPs resulted in a disempowering experience for people after stroke [56], specifically with respect to goal setting and decision-making during rehabilitation [57].

We found the process of ESD was described as both collaborative and goal oriented. Goal setting within stroke rehabilitation is considered to be "best practice" [58] and can have a positive impact on participation in the rehabilitation process and may contribute to the recovery process [59, 60]. Goal setting should be meaningful, client-centred, specific to people after stroke with strong communication by both HCPs and people after stroke [61–63]. This type of goal setting was evident in several included studies [40, 42, 46–48]. Contrasting experiences in goal setting were experienced by people after stroke in two studies [41, 54].

The transition from hospital to home was eased by the presence of ESD however continuing with rehabilitation following discharge from ESD was often challenging due to the accessibility and availability of community stroke services. Most randomised controlled trials of ESD describe ESD service delivery over a period of four weeks however some trials allocated additional time to continue the service for participants [17]. Studies included in our synthesis varied in the time period over which ESD service was provided with only one ESD service offering a service of four weeks duration of up to five days a week [53]. The length of the service provided ranged from one to four visits [50] to between four and twenty nine weeks [48] and it was unclear or not reported in four studies [41, 42, 46, 52]. Across the UK, varied and limited longer term stroke services are available following ESD [64]. Rodgers and colleagues [65] argue that the lack of evidence of effectiveness for longer term stroke interventions has led to these services not being prioritized for funding or service development in contrast with acute services. Significant gaps exist also in interdisciplinary community-based services for people after stroke in Ireland [66]. Furthermore, a synthesis of qualitative studies on people after stroke and carers experiences of primary care and community services identified issues of continuity of care and limitations in access to services [67]. Thus, longer term community-based rehabilitation services for people after stroke is an enduring problem not unique to

people discharged from ESD services and remains a challenge even where ESD extends over a longer period than the average 4 week duration.

Seven studies included in this review highlighted unmet needs or challenges of both people after stroke and carers [40–42, 44, 50, 52, 54]. These unmet needs could be considered as "life after stroke" issues which have emerged as a separate entity in recent times but yet have relatively few research studies covering the entire lifespan [68]. One important need consistently reported across studies was the need for more information and education about stroke, secondary prevention and medication for both the person after stroke and carers. These topics reflect those reported in other studies on unmet educational needs after stroke [69] and unmet information needs have been found to be the most common longer term unmet service need reported by people after stroke in surveys [70].

A Cochrane review of trials of self-management programs for people after stroke found that all interventions routinely included stroke related education including secondary prevention [71]. The review concluded that these programs may benefit the quality of life and self-efficacy of people after stroke. Of note, stroke latency varied in the included studies from one month to one year or more [71], therefore it may be worthwhile exploring effective approaches to address the information needs of people after stroke and carers in the early stage after stroke. Across the studies, carers described feeling thrown into the caring role and not supported sufficiently by the ESD service with HCPs suggesting the need to better support families. A study of carers' experiences of ESD in the UK reported that they valued being dealt with in a sensitive and thoughtful manner and having support available to them [72]. The burden experienced by carers of people after stroke has been well documented [73] and specifically the potential for home-based rehabilitation to put a high burden on carers has been identified [74]. The Cochrane review of ESD found no apparent difference in carer subjective health status scores, no apparent reduction in the mood score of carers receiving ESD services and no convincing difference in the odds of carers who received ESD services expressing satisfaction with services [17]. Of note, of the seventeen included trials in the review only nine measured carer subjective health status, four measured carer satisfaction and only three measured carer mood [17].

Carers are considered central to the rehabilitation journey [40] and should be supported in assuming the role of carer. A Cochrane review on information provision for people after stroke and their carers found active information provision for carers may reduce anxiety and depression scores slightly [75]. A subsequent systematic review found that psychosocial interventions were effective in reducing depression in carers [76]. In addition, a feasibility study of a biopsychosocial intervention for carers which combined education about stroke specific topics along with strategies focussing on successful adjustment to stroke and caregiving was determined to be suitable for carers however further work to test the delivery is needed [77, 78]. Despite recruitment challenges in the feasibility study, several carers reported they would have valued the intervention earlier in the stroke pathway. There is a need for further research to inform services to support carers of people after stroke in assuming this role and the optimal timing of these interventions particularly in the context of ESD.

Organisational and interprofessional factors were perceived by participants in this synthesis to be central to the success of ESD. These factors included the need for clarity on the ESD model of care, the need for some flexibility in the application of ESD criteria, the need for advance planning of discharge from ESD, the need for person-centred ESD scheduling and the need for ESD HCPs to work effectively as an interprofessional team. Evidence is emerging of the effectiveness of ESD with other populations such as people with chronic obstructive pulmonary disorder (COPD), and post orthopaedic surgery, respiratory infections and for the

older person [79–84]. Therefore, our findings on organisational and logistical factors that supported the success of ESD may be applicable to these services also such as the need for interprofessional practice and clarity on the model of care.

Our findings on important organisational and interprofessional factors in ESD reflect many best practice guidelines or recommendations for stroke services such as Canadian recommendations which emphasise the importance of transition planning across the stroke care pathway and the need for effective interprofessional practice [85]. Our review suggests that the transition from hospital to home was considered "seamless" [44] however people after stroke and families reported little knowledge of what ESD was before discharge home. HCPs in acute settings should ensure people after stroke and carers understand what ESD is and what it will entail before discharge to the home environment as successful transitions in the stroke care pathway should be person-centred with timely communication [86].

Identification of a key worker or key co-ordinator may become a link between the ESD team, acute hospital staff, people after stroke and carers to support care transitions and act as a key point of contact. Whilst perhaps not employed directly in the role, Fisher and colleagues [40] reported that ESD HCPs assumed this role in order to liaise with other services or to sign-post people after stroke to services. Similarly, a case manager or ESD coordinator role was described in the studies by Hitch and colleagues [53] and von Koch and colleagues [48] which highlighted that one of the HCPs was a case manager responsible for coordinating the discharge, championing the service, working across service boundaries and being the single point of contact and coordination. The potential benefits of case managers or stroke navigators in reducing health system burden through empowering people after stroke, aiding access to community resources and linkages has been identified [85] and warrants further investigation in future research.

RAs were described as a positive addition to ESD teams by HCPs in the included studies. Preliminary evidence of the effectiveness of RAs for improving some clinical and organisational outcomes has been identified in a recent systematic review [87] however no ESD studies with people after stroke were included in the review. ESD has been shown to be cost effective and cost saving [21]. Future research should evaluate the effectiveness and cost implications of the delegation of some aspects of HCPs roles in ESD to RAs.

## Strengths and limitations of the review

**Strengths of the review.**   We included 14 studies and have successfully generated an enhanced understanding of experiences of ESD. Rigorous and transparent methods were employed in the conduct and reporting of the synthesis. The broad search string and timeframe applied to the searches enhanced the rigour relating to study identification.

**Limitations.**   Several limitations were associated with this synthesis. We limited our inclusion criteria to English language studies only and included studies were all conducted in high income settings. Additional studies may have been found whereby English was not the first language it was published in which may have generated additional information. Data on carers' experiences of ESD was somewhat limited as only four of the included studies had carer involvement. Seven studies included the numbers and interviews completed with HCPs whether they were working in a clinical or managerial capacity however it was unclear in one study as to some of the disciplines interviewed [40]. Across the seven studies, only one clearly outlined that an interview was conducted with a Social Worker [49] therefore the wider experience of certain ESD team members may require further consideration. None of the included studies followed participants over time which if completed may enhance our understanding and learning of the experiences of ESD.

### Future research

This review has highlighted a number of areas that require further research. People after stroke and carers expressed concern in transitioning from ESD to community services due to the limited availability of stroke services in the community. With participant engagement; the development and evaluation of new models of care could be considered to enhance this transition. Future research should also focus on identifying effective ways to support families/carers during ESD as they assume the role of carer for the first time in some cases. Future research on the effectiveness of case managers and RAs in the ESD team is also warranted.

### Conclusion

This synthesis found that ESD is experienced as a goal-oriented and collaborative process leading to positive outcomes. ESD eased the transition from hospital to home however the transition from ESD to community services was often problematic. Healthcare providers, carers and people after stroke agreed that the home environment enhanced rehabilitation. Various organisational and interprofessional factors influenced the success of ESD. Finally, despite the provision of ESD services a number of unmet needs persisted particularly information needs and carer support needs. Future research should focus on developing and evaluating models of care to enhance the transition from ESD to community services and interventions to support carers and address information needs after stroke.

### Supporting information

**S1 File. Search string.**
(DOCX)

**S2 File. ENTREQ checklist.**
(DOCX)

**S3 File. PRISMA checklist.**
(DOCX)

### Author Contributions

**Conceptualization:** Elaine O. Connor, Rose Galvin, Katie Robinson.

**Data curation:** Elaine O. Connor, Rose Galvin, Katie Robinson.

**Formal analysis:** Elaine O. Connor, Rose Galvin, Katie Robinson.

**Methodology:** Elaine O. Connor, Rose Galvin, Katie Robinson.

**Project administration:** Elaine O. Connor.

**Supervision:** Rose Galvin, Katie Robinson.

**Writing – original draft:** Elaine O. Connor.

**Writing – review & editing:** Elaine O. Connor, Eamon Dolan, Frances Horgan, Rose Galvin, Katie Robinson.

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
