## [Decision Letter · Decision Letter 0]

4 Dec 2022

PONE-D-22-26987A qualitative evidence synthesis exploring people with stroke, family members, carers and healthcare professionals' experiences of early supported discharge (ESD) after strokePLOS ONE

Dear Dr. O Connor,

Thank you for submitting your manuscript to PLOS ONE. After careful consideration, we feel that it has merit but does not fully meet PLOS ONE’s publication criteria as it currently stands. Therefore, we invite you to submit a revised version of the manuscript that addresses the points raised during the review process.

We look forward to receiving your revised manuscript.

Kind regards,

Anuchart Kaunnil, PhD

Academic Editor

PLOS ONE

Journal Requirements:

Reviewers' comments:

Reviewer's Responses to Questions

**Comments to the Author**

1. Is the manuscript technically sound, and do the data support the conclusions?

Reviewer #1: Yes

Reviewer #2: Yes

Reviewer #3: Yes

2. Has the statistical analysis been performed appropriately and rigorously? 

Reviewer #1: N/A

Reviewer #2: N/A

Reviewer #3: No

3. Have the authors made all data underlying the findings in their manuscript fully available?

Reviewer #1: Yes

Reviewer #2: Yes

Reviewer #3: Yes

4. Is the manuscript presented in an intelligible fashion and written in standard English?

Reviewer #1: Yes

Reviewer #2: Yes

Reviewer #3: No

5. Review Comments to the Author

Reviewer #1: There are some newly published papers that related with ESD such as "Early Supported Discharge and Transitional Care Management After Stroke: A Systematic Review and Meta-Analysis". It would be better if you include these publications in your literature review or discussion.

Reviewer #2: Comment to the Author

Thank you for the opportunity to review the manuscript titled “A qualitative evidence synthesis exploring people with stroke, family members, carers and healthcare professionals' experiences of early supported discharge (ESD) after stroke”. The study is relevant to synthesize the experiences and views of early supported discharge systems after stroke from the perspective of stroke individuals, family members, carer, and healthcare professionals by systematic review of the existing qualitative research.

This manuscript was well-written and well-organized. However, minor revisions are required to make this publishable - that does not mean there are shortcomings in your work. We appreciate your contribution and hope the authors consider my feedback for enhancing their manuscript.

Introduction:

The final paragraph, the author should explain the gap from the study of Osborne and Neville in terms of the limitations of this study and how your review could strengthen those limitations.

Methods:

The design is purposeful to answer the aim of the study.

Results:

Line 204: Please provide the full version of the term “HCPs” since it is the first place that this term is appears

Overall: Please check the consistency of the abbreviation since there are HPC and HPCs appearing in the text.

Discussion:

Line 598: Please re-establish the study purpose before the main finding.

Line 603: Please provide the full name of theme 5.

Also, the discussion part mostly highlights and discusses the other themes. I found less discussion on the topic of organisational, logistical and interprofessional factors. If the authors could provide more details about how this topic is similar or different to other studies or how this topic could extend the results from previous studies it would be better.

Overall, please kindly check references and follow Vancouver style as it is required by the publisher.

Best wishes,

Reviewer #3: Dear authors,

It is my pleasure to review your manuscript on ESD among patients with stroke, their family members, carer, and health professionals. This is a good area to be further explored as it provides implications to inform clinical practice, research, and policy. However, I think that the manuscript requires some revisions to improve its quality and flow., especially in the findings section. Please consider to do proofreading for the manuscript as I found that some sentences have structural and grammatical issues. Personally, I think that this manuscript used heterogenous samples in which the conclusion of the experiences and perspectives should be cautiously made according to the sample group. Therefore, I would like you to consider the following to improve your manuscript.

Title

You used both stakeholders and ‘sample names’ in the manuscript. It would be good if it is consistent throughout the manuscript to avoid confusion.

Abstract

Please amend the abstract as you made the changes to your manuscript.

Introduction

Some sentences in the paragraph 1 and 2 require some revisions to improve the structure and clarity.

In the paragraph 2, it would be good if you could define ESD clearly. It was not clear how early is early in the context of ESD.

The gap of the study was not clear. You did state that this was an extension work from the previous one. However, what were the issues and limitations of the previous studies other than limited databases used? And what are the rationale and significance of the present review? How could it contribute to the body knowledge on the topic/area?

Methodology

It was good that you used ENTREQ as a reporting guideline for this synthesis review. However, it would be good if you could report who performed the search, whether the search was performed independently of in group.

Could you please provide justification of using CASP as a critical appraisal tools and how it helped you in the evidence synthesis process?

You state in the introduction and the methodological part that you used a three-stage approach as a method of thematic analysis. What are the good things of using this approach? And how difference it was as compared to other methods? The step in the analysis process was not clear and require revision.

Findings

I think that this part requires a major revision. I would recommend you rewrite and rearrange the themes and subthemes accordingly. The name of the themes can be simplified as it is too long. For instance, ESD ease transition to home, but not to community.

It is not clear how many subthemes under each of the themes and please have a look on the connection between themes to avoid redundancies.

I think that you also need to synthesize and explain the context of the reviewed studies as some was carried out with different samples and purposes.

Please use consistent terms throughout the manuscript. For example, patient vs people. And please introduce the full terms first before using the abbreviations. E.g. . OT/PT

Discussion

It would be good if you can start the discussion by restating the aim of the study before further discuss your main findings.

Some parts of discussion was too descriptive and lack of critical analysis. It seemed that you were just reporting previous studies’ findings. For example, the last paragraph in the discussion.

Some of the sentences in the discussion require revision to improve the clarity. For examples the last sentence in the third paragraph of discussion part.

Strength and weakness

Please state the limitations of the review.

Conclusion

The second last paragraph in the conclusion need to be rephrased for clarity and a better structure.

6. PLOS authors have the option to publish the peer review history of their article (what does this mean?). If published, this will include your full peer review and any attached files.

Reviewer #1: No

Reviewer #2: No

Reviewer #3: No

---

## [Author Response · Author response to Decision Letter 0]

24 Jan 2023

The authors wish to thank the three reviewers sincerely for their thoughtful reviews of our manuscript. Reviewer comments were carefully considered, and we have endeavoured to comprehensively address their suggestions and concerns. The table below outlines the changes we made to the manuscript in response to reviewer comments. 

 Reviewer comment Author Response

Reviewer #1 

 There are some newly published papers that related with ESD such as "Early Supported Discharge and Transitional Care Management After Stroke: A Systematic Review and Meta-Analysis". It would be better if you include these publications in your literature review or discussion. Thank you for this suggestion. This paper and another recent paper are now referred to in the introduction. 

Reviewer # 2 

 Introduction:

The final paragraph, the author should explain the gap from the study of Osborne and Neville in terms of the limitations of this study and how your review could strengthen those limitations.

 The limitations of the review by Osborne and Neville are more clearly articulated at the close of the introduction. 

 Results:

Line 204: Please provide the full version of the term “HCPs” since it is the first place that this term is appears

Overall: Please check the consistency of the abbreviation since there are HPC and HPCs appearing in the text. 

 Full version of HCP (health care provider) included on page 5.

Where we refer to a singular provider we use HCP and where we refer to multiple providers we have used HCP’s/HCPs.

 Line 598: Please re-establish the study purpose before the main finding.

 The study aim has been inserted at the outset of the discussion section. 

 Line 603: Please provide the full name of theme 5. We have amended and shortened theme titles in line with feedback from reviewer #3. 

 Also, the discussion part mostly highlights and discusses the other themes. I found less discussion on the topic of organisational, logistical and interprofessional factors. If the authors could provide more details about how this topic is similar or different to other studies or how this topic could extend the results from previous studies it would be better.

 This theme is now more comprehensively addressed in the discussion section. 

 Overall, please kindly check references and follow Vancouver style as it is required by the publisher. Thank you for this feedback all references have been reviewed and amended in line with the published required style. 

Reviewer # 3 

Title

You used both stakeholders and ‘sample names’ in the manuscript. It would be good if it is consistent throughout the manuscript to avoid confusion. This has been changed throughout to ensure consistency. 

 Abstract

Please amend the abstract as you made the changes to your manuscript. Some minor changes were made to the abstract as indicated in tracked changes in response to revisions to the manuscript. 

 Introduction

Some sentences in the paragraph 1 and 2 require some revisions to improve the structure and clarity. 

In the paragraph 2, it would be good if you could define ESD clearly. It was not clear how early is early in the context of ESD. 

Paragraph 1 and 2 have been edited to improve readability and clarity. 

The average hospital length of hospital stay in the UK based on national stroke registry data has been included to clarify how early is early in the context of ESD. 

 Introduction: 

The gap of the study was not clear. You did state that this was an extension work from the previous one. However, what were the issues and limitations of the previous studies other than limited databases used? And what are the rationale and significance of the present review? How could it contribute to the body knowledge on the topic/area? 

 At the close of the introduction we have more explicitly stated how stakeholder stroke care pathways are a priority for future research to more clearly argue the value of our synthesis. We also describe in more details the limitations of the previous review by Osborne and Neville 

 It was good that you used ENTREQ as a reporting guideline for this synthesis review. However, it would be good if you could report who performed the search, whether the search was performed independently of in group. In line with the ENTREQ guidelines, we have included who performed the searches in ‘search selection’ section page 7. 

 Could you please provide justification of using CASP as a critical appraisal tools and how it helped you in the evidence synthesis process? Choice of CASP is now justified in section ‘critical appraisal and data extraction’ and we describe how CASP scores influenced analysis. 

 You state in the introduction and the methodological part that you used a three-stage approach as a method of thematic analysis. What are the good things of using this approach? And how difference it was as compared to other methods? The step in the analysis process was not clear and require revision.

 A rationale for choosing thematic synthesis and expanded description of the synthesis process is presented on page 9. 

 I think that this part requires a major revision. I would recommend you rewrite and rearrange the themes and subthemes accordingly. The name of the themes can be simplified as it is too long. For instance, ESD ease transition to home, but not to community. 

It is not clear how many subthemes under each of the themes and please have a look on the connection between themes to avoid redundancies. Thank you for this feedback. The results section has been edited extensively to avoid redundancies / duplication across the themes. Theme titles have been simplified. 

While subthemes are not explicitly presented each theme has a number of core ideas / concepts and we have endeavoured to make these clear for the reader. The results section is now more succinct and this will also aid readability and interpretation of the findings. 

 I think that you also need to synthesize and explain the context of the reviewed studies as some was carried out with different samples and purposes.

 On page 11 an expanded description of the aims of original studies is presented and greater detail on original samples is included. 

 Please use consistent terms throughout the manuscript. For example, patient vs people. And please introduce the full terms first before using the abbreviations. E.g. . OT/PT 

 We have edited the manuscript to ensure consistency in use of terms and to ensure all terms are used in full before being abbreviated 

The use of the word patient appears only in direct text used from included articles and in Table 2 

 It would be good if you can start the discussion by restating the aim of the study before further discuss your main findings. The study aim has been inserted at the outset of the discussion section.

 Some parts of discussion was too descriptive and lack of critical analysis. It seemed that you were just reporting previous studies’ findings. For example, the last paragraph in the discussion. The discussion has been revised to include critical analysis and interpretation of findings. In line with feedback from both reviewer 2 and 3 

 Some of the sentences in the discussion require revision to improve the clarity. For examples the last sentence in the third paragraph of discussion part. The discussion has been edited to improve clarity and readability. 

 Strength and weakness

Please state the limitations of the review. On page 31 limitations of the review are outlined. 

 Conclusion

The second last paragraph in the conclusion need to be rephrased for clarity and a better structure. This paragraph has been edited for clarity. 

 Please consider to do proofreading for the manuscript as I found that some sentences have structural and grammatical issues. The manuscript has been proofread and structural and grammatical errors corrected.

---

## [Editor Report · Decision Letter 1]

26 Jan 2023

A qualitative evidence synthesis exploring people after stroke, family members, carers and healthcare professionals' experiences of early supported discharge (ESD) after stroke

PONE-D-22-26987R1

Dear Dr. O Connor,

We’re pleased to inform you that your manuscript has been judged scientifically suitable for publication and will be formally accepted for publication once it meets all outstanding technical requirements.

Kind regards,

Anuchart Kaunnil, PhD

Academic Editor

PLOS ONE
---

## [Editor Report · Acceptance letter]

3 Feb 2023

PONE-D-22-26987R1 

A qualitative evidence synthesis exploring people after stroke, family members, carers and healthcare professionals’ experiences of early supported discharge (ESD) after stroke 

Dear Dr. O Connor:

I'm pleased to inform you that your manuscript has been deemed suitable for publication in PLOS ONE. Congratulations! Your manuscript is now with our production department. 

Kind regards, 

on behalf of

Dr. Anuchart Kaunnil 

Academic Editor

PLOS ONE